# THE GUIDE AND THE EXPLORER: SMART AGENTS FOR RESOURCE-LIMITED ITERATED BATCH REINFORCEMENT LEARNING

## ABSTRACT

Iterated batch reinforcement learning (RL) is a growing subfield fueled by the demand from systems engineers for intelligent control solutions that they can apply within their technical and organizational constraints. Model-based RL (MBRL) suits this scenario well for its sample efficiency and modularity. Recent MBRL techniques combine efficient neural system models with classical planning (like model predictive control; MPC). In this paper we add two components to this classical setup. The first is a Dyna-style policy learned on the system model using model-free techniques. We call it the guide since it guides the planner. The second component is the explorer, a strategy to expand the limited knowledge of the guide during planning. Through a rigorous ablation study we show that exploration is crucial for optimal performance. We apply this approach with a DQN guide and a heating explorer to improve the state of the art of the resource-limited Acrobot benchmark system by about 10%.

## 1 INTRODUCTION

John is a telecommunication engineer. His day job is to operate a mobile antenna. He has about forty knobs to turn, in principle every five minutes, based on about a hundred external and internal system observables. His goal is to keep some performance indicators within operational limits while optimizing some others. In the evenings John dreams about using reinforcement learning (RL) to help him with his job. He knows that he cannot put an untrusted model-free agent on the antenna control (failures are very costly), but he manages to convince his boss to run live tests a couple of days every month.

John's case is arguably on the R&D table of a lot of engineering companies today. AI adoption is slow, partly because these companies have little experience with AI, but partly also because the algorithms we develop fail to address the constraints and operational requirements of these systems. What are the common attributes of these systems?

- They are physical, not getting faster with time, producing tiny data compared to what model-free RL (MFRL) algorithms require for training.
- System access is limited to a small number of relatively short live tests, each producing logs that can be used to evaluate the current policy and can be fed into the training of the next.
- They are relatively small-dimensional, and system observables were designed to support human control decisions, so there is no need to filter them or to learn representations (with the exception when the engineer uses complex images, e.g., a driver).
- Rewards are non-sparse, performance indicators come continually. Delays are possible but usually not long.
- Safe operation even during live tests is crucial. Simulators and digital twins are more and more available, but some systems are too complex to simulate from first principles.

The RL setup that fits this scenario is neither pure batch (offline; Levine et al. (2020)) as interacting with the system is possible during short periods of time nor pure online as the policy can only be

updated offline between two interaction periods. We refer to it as *iterated batch RL* (Lange et al. (2012) call it *growing* batch). Furthermore, we are interested in model-based RL (Deisenroth & Rasmussen, 2011; Chua et al., 2018; Moerland et al., 2021) because i) it is sample efficient (Chua et al., 2018; Wang et al., 2019), ii) it works well on small-dimensional systems with dense rewards, and iii) the system model (data-driven simulator) itself is an object of interest because it can ease the adoption of data-driven algorithms by systems engineers.

Given a robust system model, simple model predictive control (MPC) agents using random shooting (RS; Richards (2005); Rao (2010)) or the cross entropy method (CEM; de Boer et al. (2004)) have been shown to perform remarkably well on many benchmark systems (Nagabandi et al., 2018; Chua et al., 2018; Wang et al., 2019; Hafner et al., 2019; Kégl et al., 2021) and real-life domains such as robotics (Yang et al., 2020). On the other hand, implementing successfully the seemingly elegant Dyna-style approach (Sutton, 1991; Kurutach et al., 2018; Clavera et al., 2018; Luo et al., 2019), when we learn model-free agents on the system model and apply them on the real system, remains challenging especially on systems that require planning with long horizon. Our main finding is that the Dyna-style approach can be an excellent choice for iterated batch RL by adding to it a decision-time planning guided by the model-free policy and a dynamically self-tuning exploration. The key is that exploration and bootstrapping with a value function estimate affords us shorter rollouts so we can reduce the accumulation of errors that plagues the Dyna-style approach with long horizons (Janner et al., 2019). We also innovate on the experimental framework (metrics, statistically rigorous measurements), so we can profit from the modularity of the Dyna-style approach, tuning ingredients (model, the MFRL *guide* policy, exploration, planning, bootstrapping) independently. This modular approach makes engineering easier (as opposed to monolithic approaches like AlphaZero (Silver et al., 2017)), which is an important aspect if we want to give the methodology to non-expert systems engineers.

We demonstrate our GUIDE&EXPLORE approach on the small-dimensional but difficult Acrobot system which is known to be especially tough for Dyna-style techniques (Wang et al., 2019). We show in a rigorous ablation study how each ingredient adds a small but significant improvement, achieving a 10% total margin over the current state of the art (Wang et al., 2019; Wang & Ba, 2020; Kégl et al., 2021). DYNAZERO (pure Dyna-style with an AlphaZero guide) also matches the asymptotic performance with a larger sample complexity (3x) and computational (5x) price tag.

Table 1: Summary of previous scores and our results on resource-limited Acrobot ($n$: planning rollouts, $L$: horizon). MAR is the episodewise mean reward measured on the second half of the episodes where the algorithms have safely achieved their asymptotic performance. The two RS rows are random shooting planning *on the real system*, requiring orders of magnitudes more system access steps than MBRL. ↓ and ↑ mean lower and higher the better, respectively.

| Method | Style | Sys acc steps ↓ | MAR ↑ |
|---|---|---|---|
| POPLIN-A (Wang & Ba, 2020) | model-based MPC | 50 K | 2.116±0.010 |
| PETS-RS (Chua et al., 2018; Wang et al., 2019) | model-based MPC | 50 K | 1.908±0.010 |
| DARMDN-RS (Kégl et al., 2021) | model-based MPC | 20 K | 2.075±0.010 |
| RS($n = 100, L = 10$) | model-free MPC | 200 K | 2.106±0.047 |
| RS($n = 100\,\text{K}, L = 20$) | model-free MPC | 400 M | 2.583±0.082 |
| **GUIDE&EXPLORE**($n = 100, L = 10$) | model-based Dyna | **20 K** | **2.284**±0.012 |
| **DYNAZERO**($n = 500$) | model-based Dyna | **20 K** | **2.304**±0.031 |

## 1.1 SUMMARY OF CONTRIBUTIONS

- A conceptual framework with interchangeable algorithmic bricks for iterative batch reinforcement learning, suitable to bring intelligent control into slow, physical, low-dimensional engineering systems and the organizational constraints surrounding them.

- A rigorous experimental framework to optimize such systems.

- An ablation study that helped us find the combination of a neural model, a bootstrapping DQN guide, and a heating explorer, which lead to a 10% jump in the state of the art on the resource-limited Acrobot system.

## 2 RELATED WORK

The MBRL subfield has seen a proliferation of powerful methods, but most of them miss the specific requirements (solving problems irrelevant in this scenario like representation learning or sparse rewards) and missing others (limited and costly system access; data taking and experimentation through campaigns, live tests; safety) (Hamrick, 2019).

The Dyna framework developed by Sutton (1991) consists in training an agent from both real experience and from simulations from a system model learned from the real data. Its efficient use of system access makes it a natural candidate for iterated batch RL. The well-known limitation of this approach is the agent overfitting the imperfect system model (Grill et al., 2020). A first solution is to use short rollouts on the model to reduce error accumulation as done in Model-Based Policy Optimization (MBPO; Janner et al. (2019)). Another solution is to rely on ensembling techniques for the model. Kurutach et al. (2018)'s ME-TRPO is based on an ensemble of models and Trust Region Policy Optimization (TRPO; Schulman et al. (2015)). At each TRPO step a transition is sampled from a randomly picked model of the ensemble, preventing the policy from overfitting one model. An ensemble is also used in Model-Based Meta-Policy Optimization (MP-MPO; Clavera et al. (2018)) where each model of the ensemble is seen as a different task used to meta-learn a policy. The meta-policy is then able to quickly adapt to any of the dynamics model and is more robust to model inaccuracies. Instead of learning the model and then the policy from the model, Stochastic Lower Bound Optimization (SLBO; Luo et al. (2019)) alternates between model and policy updates. In our ITERATEDMBRL skeleton (Fig 1), this strategy would couple the LEARN and MFRL steps, which we do not study in this paper. We note that according to the results shown in Fig 1.(a) and Table 1 in Wang et al. (2019), ME-TRPO, SLBO and MB-MPO alone (pure Dyna, without planning) are clearly suboptimal on the Acrobot system[1], similar to the performance obtained by our pure Dyna DQN (Section 4). Finally, Yu et al. (2020) and Kidambi et al. (2020) use a Dyna-style approach in the context of pure batch RL where no further data collection and therefore no further model updates are assumed.

The idea of using a guide and a value function when planning is not novel (Silver et al., 2017; Schrittwieser et al., 2020; Wang & Ba, 2020; Argenson & Dulac-Arnold, 2020). We were greatly inspired by these elements in our objective of building smarter agents as they can make the search more efficient and thus lead to a better performance. POPLIN-A (Wang & Ba, 2020) relies on behavior cloning (using only real experience, unlike our Dyna-style approach that mainly uses the model), but their guide is similar to our approach. During the planning, they add random noise to the actions recommended by a deterministic policy network and update the noise distribution using a CEM strategy. In a similar way our GUIDE&EXPLORE strategy also adds a carefully controled amount of noise to the recommended actions. Our results highlight the importance of a well-calibrated exploration, which also contributes to the understanding of POPLIN-A. Argenson & Dulac-Arnold (2020) and Lowrey et al. (2019) both found that bootstrapping with a value estimate improves the performance of simple guided MPC strategies. The popular AlphaZero (Silver et al., 2017) and MuZero (Schrittwieser et al., 2020) algorithms also rely on a guide and a value function for their Monte Carlo Tree Search (MCTS). We implement a Dyna-style version of AlphaZero, which we call DYNAZERO. The principal issue of MuZero (Schrittwieser et al., 2020) in our micro-data iterated batch RL context is that it does not control the number of system access steps: it needs to simulate a lot from the *real* environment to establish the targets for the value function. In these two algorithms the guide is updated from the results obtained during the search that it guided, a procedure known as Dual Policy Iteration (Sun et al., 2018). We prefer experiencing with Dyna-style approaches first to leverage popular MFRL algorithms and defer the study of Dual Policy Iteration to future work.

Our results show that planning is an important ingredient, a claim already made by Hamrick et al. (2021). They use MuZero to run their ablation study while we prefer using an explicit model for practical reasons explained in the introduction. Besides planning we also study the importance of exploration.

---

[1]The conversion from Wang et al. (2019): our mean reward per episode of 200 steps $=$ their return$/200 + 1$.

## 3 THE FRAMEWORK FOR RESOURCE-LIMITED ITERATIVE BATCH RL

### 3.1 THE FORMAL SETUP

Let $\mathcal{T}_T = \big((\boldsymbol{s}_1, \boldsymbol{a}_1), \ldots, (\boldsymbol{s}_T, \boldsymbol{a}_T)\big)$ be a system trace consisting of $T$ steps of observable-action pairs $(\boldsymbol{s}_t, \boldsymbol{a}_t)$: given an observable $\boldsymbol{s}_t$ of the system state at time $t$, an action $\boldsymbol{a}_t$ was taken, leading to a new system state observed as $\boldsymbol{s}_{t+1}$. The observable vector $\boldsymbol{s}_t = (s_t^1, \ldots, s_t^{d_s})$ contains $d_s$ numerical or categorical variables, measured on the system at time $t$. The action vector $\boldsymbol{a}_t$ contains $d_a$ numerical or categorical action variables, typically set by a control function $\boldsymbol{a}_t = \pi(\boldsymbol{s}_t)$ of the current observable $\boldsymbol{s}_t$ (or by a stochastic policy $\boldsymbol{a}_t \sim \pi(\boldsymbol{s}_t)$; we will also use the notation $\pi : \boldsymbol{s}_t \rightsquigarrow \boldsymbol{a}_t$). The performance of the policy is measured by the reward $r_t$ which we assume to be a function of the observables $\boldsymbol{s}_t$. Given a trace $\mathcal{T}_T$ and a reward $r_t$ obtained at each step $t$, we define the mean reward as $\mathrm{R}(\mathcal{T}_T) = \frac{1}{T} \sum_{t=1}^{T} r_t$.[2] The system model $p : (\boldsymbol{s}_t, a_t) \rightsquigarrow \boldsymbol{s}_{t+1}$ can be a deterministic point predictor or a probabilistic (generative) model that, besides the point prediction $\mathbb{E}\big\{p\big(\boldsymbol{s}_{t+1}|(\boldsymbol{s}_t, a_t)\big)\big\}$, also provides information on the uncertainty of the prediction and/or to model the randomness of the system (Deisenroth & Rasmussen, 2011; Chua et al., 2018).

### 3.2 A NOTE ON TERMINOLOGY

By *model* we will consistently refer to the learned transition or system model $p$ (never to any policy). *Rollout* is the procedure of obtaining a trace $\mathcal{T}$ from an initial state $\boldsymbol{s}_1$ by alternating a model or real system $p$ and a policy $\pi$ (Fig 1). We decided to rename what Silver et al. (2017) calls the prior policy to *guide* since prior clashes with Bayesian terminology (as, e.g., Grill et al. (2020); Hamrick et al. (2021) also note), and guide expresses well that the role of this policy is to guide the search/planning. Sometimes the guide is also called the *reactive* policy (Sun et al., 2018) since it is typically an explicit function or conditional distribution $\xi : \boldsymbol{s} \rightsquigarrow a$ that can be executed or drawn from rapidly. We will call the (often implicit) policy $\pi : \boldsymbol{s} \rightsquigarrow a$ resulting from the guided plan/search the *actor* (sometimes also called the *non-reactive* policy since it takes time to simulate from the model, before each action). We will learn the guide $\xi$ using MFRL mainly on the model $p$, but, when possible, off-policy (since $\pi \neq \xi$) data (system trace $\mathcal{T}$) obtained by acting on the real system will also be used. In that sense, through the data from the real system, planning is part of training the guide. However, since from the point of view of the MFRL guide, the model $p$ *is* the world, we will not use the term planning to refer to the rollouts of the model $p$ and the guide $\xi$ when training the guide on $p$, rather we will reserve *planning* to the guided search procedure that results in acting on the real system.

### 3.3 EXPERIMENTAL SETUP: THE ITERATED BATCH MBRL

For rigorously studying and comparing algorithms and algorithmic ingredients, we need to fix not only the simulation environment but also the experimental setup. We parameterize the iterated batch RL loop (Fig 1) by four parameters:

- the number of episodes $N$,
- the number of system access steps $T$ per episode,
- the planning horizon $L$, and
- the number of generated rollouts $n$ at each planning step.

$N$ and $T$ are usually set by hard organizational constraints (number $N$ and length $T$ of live tests) that are part of the experimental setup. Our main goal is to measure the performance of our algorithms at a given (and challengingly small) number of system access steps $N \times T$. Planning happens *in silico*, and so $n$ and $L$ (and so the total number of calls $N \times T \times n \times L$ to the simulator/system model $p$) are softer constraints determined by the (physical) time between two steps and the computational resources available for the planning.

In benchmark studies, such as this paper, we argue that fixing $N$, $T$, $n$, and $L$ is important for making the problem well defined (taking some of the usual algorithmic choices out of the input of

---

[2]We use the *mean* reward (as opposed to the *total* reward, a.k.a return), since it is invariant to episode length and its unit is more meaningful to systems engineers.

$\text{ROLLOUT}(\pi, p, \boldsymbol{s}_1, T):$

| | |
|---|---|
| 1 | $\mathcal{T} \leftarrow \{\}$ |
| 2 | **for** $t \leftarrow 1$ **to** $T$: |
| 3 | $\qquad a_t \curvearrowleft \pi(\boldsymbol{s}_t)$   $\triangleright$ *draw action from policy* |
| 4 | $\qquad \mathcal{T} \leftarrow \mathcal{T} \cup (\boldsymbol{s}_t, a_t)$   $\triangleright$ *update trace* |
| 5 | $\qquad \boldsymbol{s}_{t+1} \curvearrowleft p(\boldsymbol{s}_t, a_t)$   $\triangleright$ *draw next state* |
| 6 | **return** $\mathcal{T}$ |

$\text{ITERATEDMBRL}(p_{\text{real}}, \mathcal{S}_0, \pi^{(0)}, N, T, L, n):$

| | |
|---|---|
| 1 | $\boldsymbol{s}_1 \curvearrowleft \mathcal{S}_0$   $\triangleright$ *draw initial state* |
| 2 | $\mathcal{T}^{(1)} \leftarrow \text{ROLLOUT}\left(\pi^{(0)}, p_{\text{real}}, \boldsymbol{s}_1, T\right)$   $\triangleright$ *initial random trace* |
| 3 | **for** $\tau \leftarrow 1$ **to** $N$:   $\triangleright$ *for N episodes* |
| 4 | $\qquad p^{(\tau)} \leftarrow \text{LEARN}\left(\cup_{\tau'=1}^{\tau} \mathcal{T}^{(\tau')}\right)$   $\triangleright$ *learn system model* |
| 5 | $\qquad \pi^{(\tau)} \leftarrow \text{ACTOR}\left(\pi^{(0)}, \pi^{(\tau-1)}, p^{(\tau)}, \cup_{\tau'=1}^{\tau} \mathcal{T}^{(\tau)}, L, n\right)$ |
| 6 | $\qquad \boldsymbol{s}_1 \curvearrowleft \mathcal{S}_0$   $\triangleright$ *draw initial state* |
| 7 | $\qquad \mathcal{T}^{(\tau+1)} \leftarrow \text{ROLLOUT}\left(\pi^{(\tau)}, p_{\text{real}}, \boldsymbol{s}_1, T\right)$   $\triangleright$ *episode trace* |
| 8 | **return** $\cup_{\tau=1}^{N} \mathcal{T}^{(\tau)}$ |

Figure 1: **The iterated batch MBRL loop.** $p_{\text{real}} : (\boldsymbol{s}_t, a_t) \rightsquigarrow \boldsymbol{s}_{t+1}$ is the real system (so Line 7 is what dominates the cost) and $p : (\boldsymbol{s}_t, a_t) \rightsquigarrow \boldsymbol{s}_{t+1}$ can be the real system or the system model in ROLLOUT. $\mathcal{S}_0$ is the distribution of the initial state of the real system. $\pi^{(0)} : \boldsymbol{s}_t \rightsquigarrow a_t$ is an initial (typically random) policy and in ROLLOUT $\pi : \boldsymbol{s}_t \rightsquigarrow a_t$ is any policy. $N$ is the number of episodes; $T$ is the length of the episodes; $L$ is the planning horizon and $n$ is the number of planning trajectories used by the actor policies $\pi^{(\tau)}$. $\tau = 1, \ldots, N$ is the episode index whereas $t = 1, \ldots, T$ is the system (or model) access step index. LEARN is a supervised learning (probabilistic or deterministic time-series forecasting) algorithm applied to the collected traces and ACTOR is a wrapper of the various techniques that we experiment with in this paper (Fig 2). An ACTOR typically updates $\pi^{(\tau-1)}$ using the model $p^{(\tau)}$ in an $n \times L$ planning loop, but it can also access the initial policy $\pi^{(0)}$ and the trace $\cup_{\tau'=1}^{\tau} \mathcal{T}^{(\tau')}$ collected on $p_{\text{real}}$ up to episode $\tau$.

the optimizer), affording meaningful comparison across papers and steady progress of algorithms. As in all benchmark designs, the goal is to make the problem challenging but not unsolvable. That said, we are aware that these choices may change the task and the research priorities implicitly but significantly (for example, a longer horizon $L$ will be more challenging for the model but may make the planning easier), so if the MBRL community can agree, it would make sense to carefully design several settings (quadruples $N - T - n - L$) on the same environment.

Our main operational cost is system access step so we are looking for any-time algorithms that achieve the best possible performance at any episode $\tau$. Hence, in the MBRL iteration (Fig 1), we use the same traces $\mathcal{T}^{(\tau)}$, rolled out in each iteration (Line 7), to i) update the model $p$ (Line 4) and the actor policy (Line 5) and ii) to measure the performance of the techniques (Section 3.5).

### 3.4 MODEL-BASED ACTOR POLICIES: GUIDE AND EXPLORE

Our main contribution is a Dyna-style GUIDE&EXPLORE strategy (Fig 2). The gist is to learn a guide policy $\xi$ using a model-free RL technique on the model $p$ and on the traces collected on the real system $\mathcal{T}$. It is known that the guide $\xi$, executed as an actor $\pi = \xi$ on the real system, does not work (we also confirm it in Section 4), partly because $\xi$ overfits the model (Fig 5 in Kurutach et al. (2018); Grill et al. (2020)), partly because the goal of $\pi$ is not only to exploit the traces $\mathcal{T} = \cup_{\tau=1}^{N} \mathcal{T}^{(\tau)}$ collected so far and model $p = \text{LEARN}(\mathcal{T})$, but also to collect data to self-improve $p$ and $\xi/\pi$ in the next episode $\tau$. This second reason is particular in our *iterated* batch setup: in *pure* batch RL, exploration is not an issue. We explore implicitly because of the randomness of planning (HETEROGENEOUSRS in Fig 3), but it turns out that it needs to be augmented by explicit exploration strategies. To show this, we experiment with two strategies, i) HEATING: modulating

the temperature of the guide distribution $\xi(\boldsymbol{a}|\boldsymbol{s})$, and ii) EPSGREEDY: choosing a random action with probability $\varepsilon$ (Fig 4). The novelty of our approach is that, instead of constant $T$ and $\varepsilon$, tuned as hyperparameters, we use a *set* of temperatures $[T_i]_{i=1}^n$ and probabilities $[\varepsilon_i]_{i=1}^n$ to further diversify the search and to let the planner to dynamically choose the right amount of randomization when selecting the best trace in Line 4 in Fig 3. Finally, similarly to Lowrey et al. (2019); Argenson & Dulac-Arnold (2020), we found that bootstrapping the planning with the learned value function at the end of each rollout trace (BOOTSTRAP in Fig 3) is crucial for optimizing the performance with a short horizon.

The main competitor of our GUIDE&EXPLORE actor is an actor that delegates all planning and exploration to the Monte-Carlo tree search of Silver et al. (2017)'s ALPHAZERO (Fig 4).

---

RSACTOR$\big(\pi^{(0)}, \pi^{\mathrm{prev}}, p, \mathcal{T}, L, n\big)$:

  1      **return** HETEROGENEOUSRS$\big([\pi^{(0)}]_{i=1}^n, p, L, n\big)$      ▷ *planning*

---

GUIDE&EXPLOREACTOR$\big(\pi^{(0)}, \pi^{\mathrm{prev}}, p, \mathcal{T}, L, n\big)$:

  1      $\xi \leftarrow$ MFRL$\big(\pi^{\mathrm{prev}}, p, \mathcal{T}\big)$      ▷ *guide policy ("Dyna-style")*
  2      $[\rho_i]_{i=1}^n \leftarrow$ EXPLORE$\big(\pi^{(0)}, \xi, n\big)$      ▷ *a set of guided explorer policies*
  3      **return** HETEROGENEOUSRS$\big([\rho_i]_{i=1}^n, p, L, n\big)$      ▷ *planning*

---

ALPHAZEROACTOR$\big(\pi^{(0)}, \pi^{\mathrm{prev}}, p, \mathcal{T}, L, n\big)$:

  1      **return** ALPHAZERO$\big(\pi^{(0)}, \pi^{\mathrm{prev}}, p, \mathcal{T}, L \times n\big)$

---

Figure 2: **Model-based ACTORs (policies executed on the real system).** RSACTOR is a classical random shooting planner that uses the random policy $\pi^{(0)}$ for all rollouts. Since the random policy has no value estimate, only TOTALREWARD can be used as VALUE in Fig 3/Line 3 of HETEROGENEOUSRS. GUIDE&EXPLOREACTOR first learns a Dyna-style *guide* policy $\xi$ on the model $p$ (more precisely, updates the previous guide contained in $\pi^{\mathrm{prev}}$). It can also use the traces $\mathcal{T}$ collected on the real system. It then "decorates" the guide by (possibly $n$ different) exploration strategies (Fig 4), and runs these reactive guide&explore policies $[\rho_i]_{i=1}^n$ in the HETEROGENEOUSRS planner, either using the raw TOTALREWARD or the total reward bootstrapped by the value estimate of the guide policy $\xi$ (BOOTSTRAP) in Fig 3/Line 3. ALPHAZEROACTOR calls ALPHAZERO which plans and explores internally, using Monte-Carlo tree search, with a budget of $L \times n$ simulator calls.

## 3.5 METRICS

We use two rigorously defined and measured metrics (Kégl et al., 2021) to assess the performance of the different algorithmic combinations. MAR measures the asymptotic performance after the learning has converged, and MRCP measures the convergence pace. Both can be averaged over seeds, and MAR is also an average over episodes, so we can detect statistically significant differences even when they are tiny, leading to a proper support for experimental development.

**MEAN ASYMPTOTIC REWARD (MAR).** Our measure of asymptotic performance, the mean asymptotic reward, is the mean reward $\mathrm{MR}(\tau) = \mathrm{R}\left(\mathcal{T}_T^{(\tau)}\right)$ in the second half of the episodes (after convergence; we set $N$ in such a way that the algorithms converge after less than $N/2$ episodes) $\mathrm{MAR} = \frac{2}{N} \sum_{\tau=N/2}^N \mathrm{MR}(\tau)$.

**MEAN REWARD CONVERGENCE PACE (MRCP$(\bar{r})$).** To assess the speed of convergence, we define the mean reward convergence pace MRCP$(\bar{r})$ as the number of steps needed to achieve mean reward $\bar{r}$, smoothed over a window of size 5: $\mathrm{MRCP}(\bar{r}) = T \times \arg\min_\tau \left(\frac{1}{5}\sum_{\tau'=\tau-2}^{\tau+2} \mathrm{MR}(\tau) > \bar{r}\right)$. The unit of MRCP$(\bar{r})$ is system access steps, not episodes, first to make it invariant to episode length, and second because in micro-data RL the unit of cost is a system access step. For Acrobot, we use $\bar{r} = 1.8$ in our experiments, which is roughly 70% of the best achievable mean reward.

$\textsc{TotalReward}(\mathcal{T})$:

    1        **return** $T \times \text{R}(\mathcal{T})$          $\triangleright$ *total reward (a.k.a return) of trace*

$\textsc{Bootstrap}(V, \alpha)(\mathcal{T})$:

    1        **return** $T \times \text{R}(\mathcal{T}^{(i)}) + \alpha V(\mathcal{T}^{(i)}[L, 1])$ $\triangleright$*total reward + value of last state*

$\textsc{HeterogeneousRS}([\rho_i]_{i=1}^n, p, L, n)[\boldsymbol{s}]$:

    1        **for** $i \leftarrow 1$ **to** $n$:
    2             $\mathcal{T}^{(i)} \leftarrow \textsc{Rollout}(\rho_i, p, \boldsymbol{s}, L)$       $\triangleright$ *ith roll-out trace*
    3             $V^{(i)} \leftarrow \textsc{Value}(\mathcal{T}^{(i)})$       $\triangleright$ *total reward of $\mathcal{T}^{(i)}$ or bootstrap*
    4        $i^* \leftarrow \arg\max_i V^{(i)}$       $\triangleright$ *index of the best trace*
    5        **return** $\mathcal{T}^{(i^*)}[1, 2]$       $\triangleright$ *first action of the best trace*

Figure 3: **$\textsc{Value}$ estimates on rollout traces and $\textsc{HeterogeneousRS}$: random shooting with a *set* of policies.** $\textsc{TotalReward}$ and $\textsc{Boostrap}$ are two ways to evaluate the value of a rollout trace. The latter adds the value of the last state to the total reward, according to a value estimate $V : \boldsymbol{s} \to \mathbb{R}^+$, weighted by a hyperparameter $\alpha$. They are called in Line 3 of $\textsc{HeterogeneousRS}$ which is a random shooting planner that accepts $n$ different shooting policies $[\rho_i]_{i=1}^n$ for the $n$ rollouts used in the search. As usual, it returns the first action $\boldsymbol{a}_1^* = \mathcal{T}^{(i^*)}[1, 2]$ of the best trace $\mathcal{T}^{(i^*)} = ((\boldsymbol{s}_1^*, \boldsymbol{a}_1^*), \ldots, (\boldsymbol{s}_T^*, \boldsymbol{a}_T^*))$. Its parameters are the shooting policies $[\rho_i]_{i=1}^n$, the model $p$, and the number $n$ and length $L$ of rollouts, but to properly define it, we also need the state $\boldsymbol{s}$ which we plan from, so we use a double argument list $()[]$.

$\textsc{HeatingExplore}(\pi^{(0)}, \xi, n)[\boldsymbol{s}]$:

    1        **for** $i \leftarrow 1$ **to** $n$:
    2             $\rho_i(\boldsymbol{a}|\boldsymbol{s}) = \dfrac{\xi(\boldsymbol{a}|\boldsymbol{s})^{1/T_i}}{\sum_{\boldsymbol{a}'} \xi(\boldsymbol{a}'|\boldsymbol{s})^{1/T_i}}$
    3        **return** $[\rho_i]_{i=1}^n$

$\textsc{EpsGreedyExplore}(\pi^{(0)}, \xi, n)[\boldsymbol{s}]$:

    1        **for** $i \leftarrow 1$ **to** $n$:
    2             $\rho_i(\boldsymbol{a}|\boldsymbol{s}) = \begin{cases} \arg\max_a \xi(\boldsymbol{a}|\boldsymbol{s}) & \text{with probability } (1 - \varepsilon_i), \\ \pi^{(0)}(\boldsymbol{a}) & \text{with probability } \varepsilon_i. \end{cases}$
    3        **return** $[\rho_i]_{i=1}^n$

Figure 4: **Exploration strategies.** $\textsc{HeatingExplore}$ heats the guide action distribution $\xi(\boldsymbol{a}|\boldsymbol{s})$ to $n$ different temperatures, and $\textsc{EpsGreedyExplore}$ changes the best action to a random action $\pi^{(0)} \rightsquigarrow \boldsymbol{a}$ with different probabilities. The temperatures $[T_i]_{i=1}^n$ and probabilities $[\varepsilon_i]_{i=1}^n$ are hyperparameters.

## 4   Experiments

### 4.1   The Acrobot benchmark environment

Acrobot is an underactuated double pendulum with four observables $\boldsymbol{s}_t = [\theta_1, \theta_2, \dot{\theta}_1, \dot{\theta}_2]$ which are usually augmented to six by taking sine and cosine of the angles (Brockman et al., 2016); $\theta_1$ is the angle to the vertical axis of the upper link; $\theta_2$ is the angle of the lower link relative to the upper link, both being clipped to $[-\pi, \pi]$; $\dot{\theta}_1$ and $\dot{\theta}_2$ are the corresponding angular momenta. For the starting position $\boldsymbol{s}_1$ of each episode, all four state variables are sampled uniformly from an approximately hanging and stationary position $s_1^j \in [-0.1, 0.1]$. The action is a discrete torque on the lower link $a \in \{-1, 0, 1\}$. The reward is the height of the tip of the lower link over the hanging position $r(\boldsymbol{s}) = 2 - \cos\theta_1 - \cos(\theta_1 + \theta_2) \in [0, 4]$.[3]

---

[3]We chose this rather than the sparse variable-episode-length version $r(\boldsymbol{s}) = \mathbb{I}\{2 - \cos\theta_1 - \cos(\theta_1 + \theta_2) > 3\}$ (Sutton, 1996) since it corresponds better to the continuous aspect of engineering systems.

Acrobot is a small but relatively difficult and fascinating system, so it is an ideal benchmark for continuous-reward engineering systems. Similarly to Kégl et al. (2021), we set the number of episodes to $N = 100$, the number of steps per episode to $T = 200$, the number of planning rollouts to $n = 100$, and the horizon to $L = 10$. With these settings, we can identify four distinctively different regimes (see the attached videos): i) the random uniform policy $\pi^{(0)}$ achieves MAR $\approx 0.1 - 0.2$ (Acrobot keeps approximately hanging), ii) reasonable models with random shooting or pure Dyna-style controllers achieve MAR $\approx 1.4 - 1.6$ (Acrobot gains energy but moves its limb quite uncontrollably), iii) random shooting $n = 100, L = 10$ with good models such as PETS (Chua et al., 2018; Wang et al., 2019) or DARMDN (Kégl et al., 2021) keep the limb up and manage to have its tip above horizon on average MAR $\approx 2.0 - 2.1$ (previous state of the art), and iv) in our experiments we could achieve a quasi perfect policy (Acrobot moves up like a gymnast and stays balanced at the top) MAR $\approx 2.7 - 2.8$ using random shooting with $n = 100\,\mathrm{K}, L = 20$ on the *real system*, giving us a target and a possibly large margin of improvement. Acrobot is also an ideal benchmark for making our point since it turned out quite challenging for Dyna-style techniques in Wang et al. (2019)'s benchmarks (MAR $\approx 1.6 - 1.7$).

## 4.2 MODELS, GUIDES, AND ACTORS

Following Kégl et al. (2021), we tried different system models (Fig 1/Line 4) from the family of Deep Autoregressive Mixture Density Networks (DARMDN) and decided to use DARMDN(1)$_{\mathrm{det}}$, an autoregressive network trained probabilistically (the output of the net for each observable dimension is a single Gaussian learned by minimizing the negative log-likelihood loss) and sampled deterministically (the mean of the Gaussian). In Kégl et al. (2021) this was the best model using random shooting; Table 4 in the Appendix confirms this using our best actor.

In principle, any MFRL technique working with discrete action space and providing a value function and a policy can be used as a guide $\xi$ when applying ITERATEDMBRL to the Acrobot system (Fig 2/Line 1). We experimented with Deep Q Networks (DQN; Mnih et al. (2015)) and PPO (Schulman et al., 2017). For the DQN, following Janner et al. (2019), short rollouts starting from real observations are performed on the model to sample transitions which are then placed in an experience replay buffer, along with the real transitions observed during the rollouts (Fig 1/Line 7). The DQN is then updated by sampling batches from this buffer. For DQN alone (without planning), an $\varepsilon$-greedy strategy is used for its training ($\varepsilon$ decaying as more training samples are used) and at decision time ($\varepsilon = 0.05$).

For actors (Fig 1/Line 5, Fig 2), we start from the simple DQN($\varepsilon = 0.05$) or PPO guide which is interacting with the system without planning. We then add different ingredients to these simple agents to improve the performance. Adding $(n, L)$ to the name of the agent means that we use the agent to guide a planning of $n$ rollouts with horizon $L$. It is important to note here that planning without exploration using the greedy guide is, in our case, equivalent to no planning since both the model $p$ and the guides $\xi$ are deterministic. XXX-HEATINGEXPLORE and XXX-EPSGREEDYEXPLORE refer to the additional use of the associated exploration strategies (Fig 1/Line 2, Fig 4). When a fixed $\varepsilon$ is used for the exploration strategy, we add it as an explicit parameter, e.g., EPSGREEDYEXPLORE($\varepsilon$). No parameters means that a different $\varepsilon$ or temperature is used for each of the $n$ rollouts. By default the value estimate is the TOTALREWARD (Fig 3). We add BOOTSTRAP when we use bootstrapping by the value function estimate. Finally, we also apply AlphaZero (Silver et al., 2017) in a Dyna-style fashion (Fig 2) which we refer to as DYNAZERO.

Appendix A contains detailed information on the various algorithmic choices.

## 4.3 RESULTS

Table 2 and Figure 5 present our the results with the main actors. We see that planning with exploration and bootstrapping gives the best results among all the DQN actors. The approach beats the previous state of the art: both RSACTOR($n = 100, L = 10$) from Kégl et al. (2021) and POPLIN-A (Wang & Ba (2020); Table 1) by 10% in terms of MAR. PPO is slightly suboptimal compared to DQN. DYNAZERO matches the best MAR but converges three times slower and takes about five times more computational time, mainly because, unlike random shooting, MCTS cannot easily be parallelized.

Table 2 and the detailed ablation study in Appendix B show that *all* ingredients add to the performance. Although DQN alone (DQN($\varepsilon = 0.05$)) is far better than a random policy, it is clearly not

sufficient to obtain a reasonable actor. Using it as a planning guide (DQN($n = 100$, $L = 10$)-EPSGREEDYEXPLORE($\varepsilon = 0.05$)) significantly improves the performance but remains worse than random shooting RSACTOR($n = 100$, $L = 10$). To propel the Dyna-style actor above simple MPC, we need to fine tune exploration. We found that allowing the planner to choose the right amount is a robust and safe approach. The final improvement was attained by adding value bootstrapping to the heated explorer/planner; bootstrapping does not improve the $\varepsilon$-greedy actor, possibly because the value estimate of the guide $\xi$ is off on random actions, unlike in the heating exploration when the explorer "softly" increases the probability of actions, given by the guide. As a reference, we also include a simple but costly RSACTOR($n = 100$ K, $L = 10$). It would not be officially accepted in our benchmark as we restrict $n$ to 100 and $L$ to 10, but it can serve as a target to reach.

Table 2: Agent evaluation results. MAR is the Mean Asymptotic Reward showing the asymptotic performance of the agent and MRCP(1.8) is the Mean Reward Convergence Pace showing the sample-efficiency (the number of system access steps required to achieve a mean reward of 1.8). ↓ and ↑ mean lower and higher the better, respectively. Except for DYNAZERO all the agents were run for 10 seeds with the ± giving the 90% confidence interval.

| Agent | MAR ↑ | MRCP(1.8) ↓ |
|---|---|---|
| RSACTOR($n = 100$ K, $L = 20$) | 2.474±0.022 | 2280.0±580.0 |
| RSACTOR($n = 100$, $L = 10$) | 2.075±0.01 | 2620.0±320.0 |
| DQN($\varepsilon = 0.05$) | 1.442±0.014 | NaN  ±NaN |
| DQN($n = 100$, $L = 10$)-EPSGREEDYEXPLORE($\varepsilon = 0.05$) | 1.932±0.012 | 3540.0±520.0 |
| DQN($n = 100$, $L = 10$)-HEATINGEXPLORE | 2.196±0.014 | 1900.0±180.0 |
| DQN($n = 100$, $L = 10$)-EPSGREEDYEXPLORE | 2.203±0.012 | 1880.0±100.0 |
| **DQN($n = 100$, $L = 10$)-HEATINGEXPLORE-BOOTSTRAP** | **2.283**±0.012 | 2180.0±280.0 |
| PPO | 1.578±0.03 | NaN  ±NaN |
| PPO($n = 100$, $L = 10$)-HEATINGEXPLORE | 2.161±0.019 | 2520.0±600.0 |
| **DYNAZERO** | **2.304**±0.031 | 6000.0±NaN |

Figure 5: Learning curves obtained with different agents. Mean reward is between 0 (hanging) and 4 (standing up). Episode length is $T = 200$, number of epochs is $N = 100$ with one episode per epoch. Except for DYNAZERO, mean reward curves are averaged across ten seeds. Areas with lighter colors show the 90% confidence intervals.

## 5  CONCLUSION

In this paper we show that an offline Dyna-style approach can be successfully applied on the Acrobot system where previously Dyna-style algorithms were failing. Our empirical results exhibit the importance of using a planning guided by a policy with the correct amount of exploration. We thus propose a strategy to automatically fine-tune exploration and add bootstrapping with a value function estimate to further improves the performance. This combination leads to an improvement over the previous state of the art by 10% while respecting the same resource constraints. Future work includes modelling the uncertainties of the value estimates so as to use them for better exploration.

## 6 REPRODUCIBILITY STATEMENT

All the code to produce our results will be made publicly available after publication. Details for the implementations of ITERATEDMBRL, the different exploration strategies, the system models, and the different MFRL agents are also provided in Section A.

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

# A    IMPLEMENTATION DETAILS

## A.1    CODE AND DEPENDENCIES

Our code will be made publicly available after publication to ease the reproducibility of all our results. We use Pytorch (Paszke et al., 2019) to build and train the neural network system models and policies. To run the ITERATEDMBRL experiments we use the `rl_simulator` (`https://github.com/ramp-kits/rl_simulator`) python library developed by Kégl et al. (2021) which relies on Open AI Gym (Brockman et al., 2016) for the Acrobot dynamics. For the PPO agent we use the StableBaselines3 implementation (Raffin et al., 2019).

## A.2    MODELS AND AGENTS

It is known that carefully tuning hyperparameters of deep reinforcement learning algorithms is crucial for success and fair comparisons (Henderson et al., 2018; Zhang et al., 2021). To reduce the computational cost and consider a reasonable search space the models and the agents were optimized independently. For the systems models we use the same hyperparameters as the ones used in Kégl et al. (2021). Please refer to Appendix D in Kégl et al. (2021) for a complete description of the hyperparameter search and the selected hyperparameters.

The DQN is a neural network with one hidden layer of 64 neurons. The input and hidden layers are both made of a linear layer followed by the ReLU activation function. The output layer is a linear layer. At the start of each epoch of ITERATEDMBRL, the system model is trained using all the data collected at the previous epochs and the DQN is updated offline from both transitions generated with the model and the real observed transitions. Following Janner et al. (2019) and Holland et al. (2019), short rollouts starting from real observations are performed on the model to sample transitions which are then placed in an experience replay buffer with total size 25,000. The real transitions observed during the rollouts on the real system are also placed in this buffer. We train and act with an $\varepsilon$-greedy strategy with an $\varepsilon$ decaying as we see more samples. The decay schedule is given by $\max(0.6 - 0.001375t, 0.05)$ where $t$ is the current number of steps realized on both the model and the real system. The DQN is updated by sampling batches of size 64 from this buffer. To generate transitions from the model we use a total number of 1800 steps. Each rollout starts from a randomly sampled real observation and performs 50 steps before being reset to a new real observation. The DQN is updated after each two steps. We use a target network which is updated after every 400 steps. Finally the learning rate is set to 0.0001 and the discount factor to 1. To obtain these values for the different hyperparameters, we started from default values and carefully tuned them to achieve the best performance.

The PPO agent is also updated offline at the start of each epoch with the model. We rely on the StableBaselines3 (Raffin et al., 2019) implementation using the default Multi-Layer Perceptron policy. A total of 15000 steps are generated and each rollout resets from a real observation each 20 steps.

For DYNAZERO we modified the version of AlphaZero available at `https://github.com/tmoer/alphazero_singleplayer`. For the policy-value network we use two hidden layers with 64 neurons each. The input and hidden layers are both made of a linear layer followed by the ELU activation function. The output layer for the policy is a softmax layer and a linear layer for the value. The constant $c$ in the UCT rule is set to 1.5 and the discount factor to 0.9. The policy-value network is updated at the beginning of each epoch by performing rollouts on the model, reset after each 50 steps. The total number of steps performed on the model at each update is 400. Data are put in a buffer of size 10,000. The batch size is set to 64 and the learning rate to 0.001. Finally the number of MCTS searches at each step is set to 500.

## A.3    EXPLORATION

For the multi-$\varepsilon$ exploration strategy based on EPSGREEDYEXPLORE we use one $\varepsilon$ value for each of the $n = 100$ rollouts: $\{0.001, 0.01, 0.02, \ldots, 0.99\}$. For the multi-temperature HEATING-EXPLORE strategy, we first normalized the $Q$ values by their maximum value, $\tilde{Q}(s, a) \; = $

$Q(s, a) / \max_{a'} Q(s, a')$, before applying a softmax:

$$\rho_i(a|s_t) = \frac{e^{\tilde{Q}(a,s_t)/T_i}}{\sum_{a'} e^{\tilde{Q}(a',s_t)/T_i}}$$

where $\{T_i, 1 \leq i \leq n\}$ is an increasing sequence of temperatures. A large temperature gives a uniform distribution, whereas a low temperature corresponds to taking $\arg\max Q(a, s_t)$. Different shapes of sequences were tried (linear, logarithmic, exponential, polynomial, logistic), and best performance was obtained with a logistic schedule (with a linear end). The exact values will be provided in the code.

## A.4 BOOTSTRAPPING

For DQN the value estimate is computed by taking the maximum of the Q values over the actions. The $\alpha$ parameter is set to $0.6$.

# B ABLATION STUDY

## B.1 ACTORS

We show in this section that a guided planning and exploration are needed to obtain the best result. Remove either planning or exploration and you will obtain suboptimal performance. To support this claim we ran an ablation study with EPSGREEDYEXPLORE. EPSGREEDYEXPLORE makes it easy to control the degree of exploration through the $\varepsilon$ parameter. Setting $\varepsilon$ to 0 corresponds to no exploration and is equivalent to using the guide greedily without planning ($n = 1$ and $L = 1$) as our model is used deterministically when sampled from. Setting $\varepsilon$ to 1 corresponds to full exploration and is equivalent to the purely random RSACTOR($n = 100$, $L = 10$). Figure 6 shows the results obtained by DQN($\varepsilon = 0.05$), DQN with a forced $\varepsilon$-greedy scheme at decision time with $\varepsilon = 0.4$ and no planning (DQN($\varepsilon = 0.4$)), and DQN with planning and fixed $\varepsilon$ values (DQN($n = 100$, $L = 10$)-EPSGREEDYEXPLORE($\varepsilon$) for $\varepsilon \in \{0.0001, 0.01, 0.05, 0.1, 0.2, 0.4, 0.8, 0.99, 0.9999\}$). All the results are also put in Table 3.

First, we see that planning definitely helps when comparing the performance of DQN($\varepsilon = 0.4$) which uses no planning but forces the exploration with $\varepsilon = 0.4$ and the performance of DQN($n = 100$, $L = 10$)-EPSGREEDYEXPLORE($\varepsilon = 0.4$) which uses planning and the same fixed value of $\varepsilon$. Second, as expected, the closer $\varepsilon$ is to 0 the closer the performance is to DQN($\varepsilon = 0.05$), and the closer $\varepsilon$ is to 1 the closer the performance is to RSACTOR($n = 100$, $L = 10$). With a well-chosen $\varepsilon$ between these two extremes, say $\varepsilon = 0.4$, we obtain a better performance than either extremes.

We can thus claim that planning is required and planning with a correct amount of exploration. Our EPSGREEDYEXPLORE or HEATEDEXPLORE exploration strategies, used with multiple $\varepsilon$ or temperatures values allows for the automatic and dynamic selection of the good amount of exploration.

## B.2 MODELS

In Table 4 we compare the performances of models from Kégl et al. (2021) when applied with our best agent DQN($n = 100$, $L = 10$)-HEATINGEXPLORE-BOOTSTRAP. Most models belong to the family of Deep (Autoregressive) Mixture Density Networks with $D$ components, D{AR}MDN($D$){det}, trained by minimizing the negative log-likelihood. The 'det' suffix means that the model is sampled from deterministically, returning the mean of the predicted distribution. DARNNs are Deep Autoregressive Neural Networks point-estimating a mean by minimizing the mean squared error and estimating a constant variance using the residual errors. These models are homoscedastic with a variance independent of the input. The reader can also refer to Kégl et al. (2021) for a complete description of these models. In the paper we use the second best model DARMDN($1$)$_{det}$ over the non-autoregressive DMDN($1$)$_{det}$ because we find autoregressive models to be easier to use in practice on real-life projects due to their ability to model dependence between the observable dimensions and to model different feature types. We also note that, as observed in Kégl et al. (2021) with RSACTOR($n = 100$, $L = 10$), the deterministic versions perform slightly better than the stochastic ones, in general.

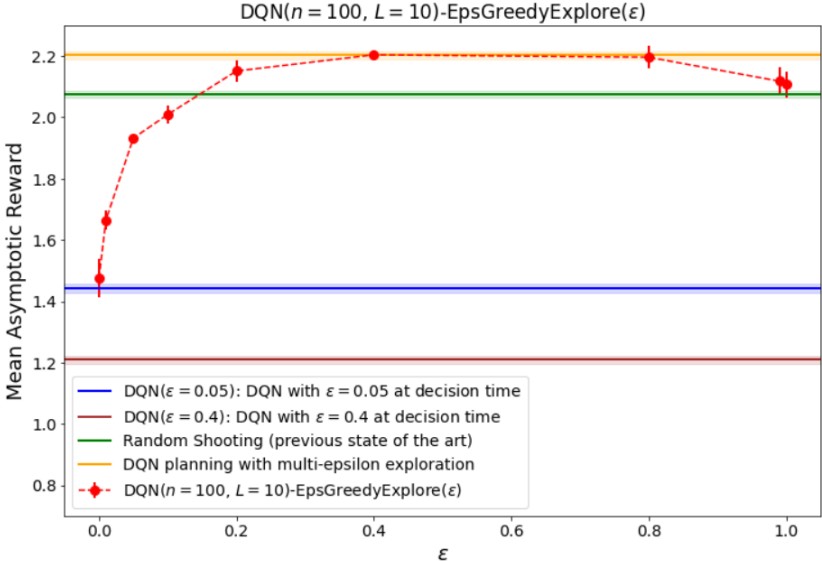

Figure 6: Mean asymptotic rewards (MAR) of DQN($n = 100$, $L = 10$)-EPSGREEDYEXPLORE($\varepsilon$) obtained for different values of $\varepsilon$. Error bars and areas in lighter colors represent the 90% confidence intervals. When $\varepsilon$ is close to 0 the performance is close to DQN while for $\varepsilon$ close to 1 the performance is close to the RSACTOR agent. Our multi-$\varepsilon$ exploration strategy is able to select the best $\varepsilon$ automatically and dynamically.

Table 3: Importance of planning and exploration. MAR is the Mean Asymptotic Reward showing the asymptotic performance of the agent and MRCP(1.8) is the Mean Reward Convergence Space showing the sample-efficiency performance as the number of system access steps required to achieve a reward of 1.8. ↓ and ↑ mean lower and higher the better, respectively. The ± values are 90% Gaussian confidence intervals.

| Agent | MAR ↑ | MRCP(1.8) ↓ | |
|---|---|---|---|
| DQN($\varepsilon = 0.4$) | 1.209±0.013 | NaN | ±NaN |
| DQN($\varepsilon = 0.05$) | 1.442±0.014 | NaN | ±NaN |
| DQN($n = 100$, $L = 10$)-EPSGREEDYEXPLORE($\varepsilon = 0.0001$) | 1.475±0.062 | NaN | ±NaN |
| DQN($n = 100$, $L = 10$)-EPSGREEDYEXPLORE($\varepsilon = 0.01$) | 1.664±0.032 | NaN | ±NaN |
| DQN($n = 100$, $L = 10$)-EPSGREEDYEXPLORE($\varepsilon = 0.05$) | 1.932±0.012 | 3540.0±520.0 | |
| DQN($n = 100$, $L = 10$)-EPSGREEDYEXPLORE($\varepsilon = 0.1$) | 2.009±0.031 | 2400.0±– | |
| RSACTOR($n = 100$, $L = 10$) | 2.075±0.01 | 2620.0±320.0 | |
| DQN($n = 100$, $L = 10$)-EPSGREEDYEXPLORE($\varepsilon = 0.9999$) | 2.107±0.042 | 2000.0±– | |
| DQN($n = 100$, $L = 10$)-EPSGREEDYEXPLORE($\varepsilon = 0.99$) | 2.118±0.046 | 2400.0±– | |
| DQN($n = 100$, $L = 10$)-EPSGREEDYEXPLORE($\varepsilon = 0.2$) | 2.151±0.034 | 2400.0±– | |
| DQN($n = 100$, $L = 10$)-EPSGREEDYEXPLORE($\varepsilon = 0.8$) | 2.196±0.037 | 2000.0±– | |
| DQN($n = 100$, $L = 10$)-EPSGREEDYEXPLORE($\varepsilon = 0.4$) | 2.204±0.01 | 1910.0±140.0 | |

## C RSACTOR PERFORMANCE ON THE REAL SYSTEM

We present the results one can obtain on the real system with an RSACTOR and different values of the planning horizon $L$ and the number of generated rollouts $n$ in Fig 7. For the considered planning horizons a larger number of generated rollouts lead to a better performance. We also observed in our simulations that for the Acrobot to stay balanced, it was necessary (although not always sufficient) to have a reward larger than 2.6. We see from Fig 7 that this can be achieved with a simple agent such as RSACTOR but at the price of a very large number of generated rollouts. The goal is therefore to design a smarter agent that can come as close as possible to this performance with a limited budget.

Table 4: Model evaluation results with for our best guide&explore strategy DQN($n = 100$, $L = 10$)-HEATINGEXPLORE-BOOTSTRAP. MAR is the Mean Asymptotic Reward showing the asymptotic performance of the agent and MRCP(1.8) is the Mean Reward Convergence Space showing the sample-efficiency performance as the number of system access steps required to achieve a reward of 1.8. ↓ and ↑ mean lower and higher the better, respectively. The performances are computed from 10 random repetitions of ITERATEDMBRL.

| Model | MAR ↑ | MRCP(1.8) ↓ |
|---|---|---|
| DMDN(1)$_\text{det}$ | 2.308±0.011 | 1860.0±240.0 |
| DARMDN(1)$_\text{det}$ | 2.297±0.012 | 2180.0±170.0 |
| DARMDN(1) | 2.294±0.012 | 2060.0±190.0 |
| DMDN(1) | 2.271±0.011 | 2120.0±400.0 |
| DARMDN(10) | 2.264±0.011 | 2840.0±720.0 |
| DARMDN(10)$_\text{det}$ | 2.244±0.013 | 1960.0±160.0 |
| DARNN$_\text{det}$ | 2.222±0.011 | 2640.0±510.0 |
| DARNN$_\sigma$ | 2.177±0.011 | 3380.0±430.0 |
| DMDN(10) | 2.148±0.015 | 5080.0±1070.0 |

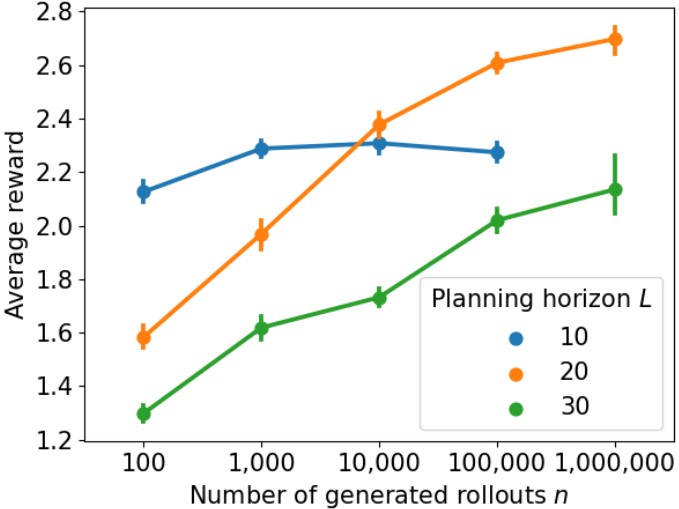

Figure 7: Performance obtained with RSACTOR on the real Acrobot system for different planning horizons $L$ and number of generated rollouts $n$. The plot shows the mean rewards obtained for several randomly initialized episodes of 200 steps. The error bars give the associated 90% confidence intervals. Note that since Acrobot has a discrete action space with three actions, the total number of different rollouts for $h = 10$ is $n = 3^{10} = 59,049$. The performance shown for $h = 10$ and $n = 100,000$ thus only requires $n = 59,049$ rollouts.

