# OpenReview forum: "The guide and the explorer: smart agents for resource-limited iterated batch reinforcement learning"
_ICLR.cc/2022/Conference — ICLR 2022 Submitted_

### Official Review · Reviewer_EmNA · 2021-10-29

**Correctness:** 1
**Technical Novelty And Significance:** 2
**Empirical Novelty And Significance:** 2
**Recommendation:** 3
**Confidence:** 4

**Main Review:**

The paper empirically shows that the proposed changes involving exploration, bootstrapped returns, and other additions improve performance of dyna algorithms in Acrobot.  However, this seems like a limited result as currently presented.  Why were these extensions not studies in other environments, especially more realistic ones like those described in the introduction?  The paper spends a lot of space describing hypothetical industrial applications but never presents domains of that kind.  In addition, the changes proposed here, especially the automated exploration techniques are not theoretically investigated.  Are there any guarantees about the convergence of Dyna+these extensions in the tabular case?

The main contribution of the paper are the empirical results on Acrobot.  But in the main paper, only a few algorithms are actually shown in Table 2 and Figure 5 (for instance where are the XXX variants?).  And the overall results in Figure 5 show virtually no difference in the asymptotic behavior of the different variants and only a small difference in convergence speed, which seems within the margin of error for the two planning variants and even comparing to current state of the art baseline (purple) there seems to be overlap by the error bars shown. I also question the use of a single percentage (70% here) in calculating the MRCP.  Ideally one would calculate convergence rate for several values, say 70%, 80%, and 90%, to make sure that convergence is proceeding at a fast pace throughout the learning process.  With only a single domain studied and marginal gains I think more studies are needed here.

The paper also does not provide any theoretical analysis of how the proposed changes will affect convergence.  Does the new “heating” exploration guarantee the optimal policy is preserved?  Is it guaranteed to converge in the limit?  What is the expected sample complexity of using this approach?

And what about the approach described on Page 6 where the planner “dynamically chooses the right amount of randomization” for exploration?  How is that actually implemented?  Is the randomization part of the available action space?  What kind of convergence guarantees are available there?  Page 9 claims that the paper shows “allowing the planner to choose the right amount is a robust and safe approach” but I do not really see evidence of that nor a concrete algorithmic description of how that procedure operates.
Lack of theory and concrete description of the hyper-parameter-less exploration

Organizationally, the paper seems unfocussed. The first page is spent on hypothetical business cases that the paper never returns to.  The related work section lists many different dyna-style algorithms but it is hard to find most of them in the actual algorithm results. I also found the pseudocode blocks quite hard to follow when comparing the different algorithm variants.  A better way to convey this information would be using a matrix where each row has an algorithm name and the columns describe the exploration type, the planner used, etc.


**Summary Of The Paper:**

The paper presents extensions and modular additions to dyna-style algorithms, especially involving bootstrapping and improved exploration.  These changes are studies on the classical, but difficult, Acrobot domain and it is shown empirically that the changes improve convergence speed and asymptotic performance.

**Summary Of The Review:**

The new algorithm is interesting and seems to make some improvements over some of the existing approaches, but the gains seem marginal, some variants re not shown in the main results, and the reported statistics could use improvement.  there is also no theoretical justification or proofs provided for these changes,

---

> ### Author Response · Authors · 2021-11-22
> **Response to reviewer EmNA**
>
> We thank the reviewer for his comments on our paper.
>
> **Industrial example:**
> The industrial example motivates the need to study the iterated batch RL setting and the development of algorithms for this setting.
>
> **Lack of theoretical guarantees:**
> Although we agree that developing theoretical guarantees would be a nice addition, it was not the point of our paper to study these kind of guarantees but rather to perform an experimental study evaluating the importance of the different components.
>
>
> **And the overall results in Figure 5 show virtually no difference in the asymptotic behavior of the different variants and only a small difference in convergence speed, which seems within the margin of error for the two planning variants and even comparing to current state of the art baseline (purple) there seems to be overlap by the error bars shown**
>
>  "Eying" learning curves is one practice that plagues reinforcement learning. The tables show that the differences are statistically significant, and that is a more important information when the goal is to incrementally improve algorithms through many ablation steps.
>
> **the planner “dynamically chooses the right amount of randomization” for exploration?  How is that actually implemented?  Is the randomization part of the available action space?**
>
> Exploration is added to the learned model-free policy as shown in step 2 of the GUIDE&EXPLOREACTOR pseudo-code in figure 2. The different exploration strategies are presented in Figure 4. During the decision-time planning we generate n trajectories by following the guide policy. The guide uses a different amount of exploration for each of the n trajectories: we have different epsilon values for the epsilon greedy strategy and different temperatures when heating the model-free policy distribution. At the end the best trajectory with its corresponding amount of exploration is selected. This is why we talk about a "dynamic selection of the amount of randomization/exploration".
>
> **The related work section lists many different dyna-style algorithms but it is hard to find most of them in the actual algorithm results.**
>
> We could indeed have incorporated an ensemble for the model to try strategies such as ME-TRPO or MBPO. However Kegl et al (2021) showed that using the PETS ensemble model (Chua et al., 2018) did not provide better performance with RSActor.

---

### Official Review · Reviewer_jVBs · 2021-11-02

**Correctness:** 3
**Technical Novelty And Significance:** 3
**Empirical Novelty And Significance:** 2
**Recommendation:** 5
**Confidence:** 4

**Main Review:**

The studied problem on an approach to sample-efficient MBRL is clearly a very significant problem in modern research on RL. The authors have nice ideas, they enhance the traditional MBRL approach with two models (called the guide and the explorer), the guide is based on a Dyna-style policy, applying of which is known to be challenging in a long horizon setting. The paper is nicely written and easy to follow. I think that the designed methodology has a significant potential of being useful in the research on MBRL, but only after the presentation and experimental evaluation are mature enough.

I would like that the researchers address my concerns listed below and related mostly to correct mathematical formalization of results, and validating the methodology in a more broad set of benchmarks and environments.

### 1. Lack of proper formalization of the setting.
After reading the studied problem statement in the introduction, it is still unclear for me what precisely can be characterized using the term ‘iterated batch RL’ coined by the authors. In particular, why it is necessary to define and name an MBRL study subgenre, rather than use the existing framework of online/offline MBRL.
I appreciate the informal and visual presentation of the studied setting using the example of John, a telecommunication engineer. However, I would appreciate a formal statement and assumptions behind the studied iterated batch RL setting. Especially that it has been studied in the literature under different names. This critique extends further, in my impression the paper lacks mathematical rigor necessary in the setting of such a problem like acrobot (authors showing) being extremely sensitive to: the formulation of observation vector (sincos vs pure angles), initial condition (instability and chaotic behavior), employed model (probabilistic / multivariate / stochastic /deterministic).

### 2. Disputable significance of the reported improvement over the state-of-the-art in a single benchmark environment
One of the main selling points of the paper is the 10% improvement for the resource-limited acrobot (in terms of Mean Asymptotic Reward after the fixed number of system access steps. Basing the paper contribution on an improvement like that for a  single benchmark can be misleading. First, the issue of ensuring a sufficient number of runs with different random seeds seems to be addressed by averaging the result from 10 different runs (standard in RL literature) and reporting confidence intervals. However, the 10% improvement may be the result of a simple hyperparameter optimization.  It is now known that exhaustive hyperparameter optimization in MBRL may result in significant performance gains, and even solving for exploitative solutions, see B. Zhang et. al. _“On the Importance of Hyperparameter Optimization for Model-based Reinforcement Learning”_ AISTATS 2021.  Therefore, it is not unexpected that an exhaustive hyper-parameter search could significantly boost the MAR/MRCP scores of the other evaluated algorithms and change the final leaderboard. Table 3 given in the appendix indicates that the authors did not, in fact, perform an exhaustive hyperparameter search. Another concern is related to the fact that the DynaZero benchmark algorithm achieves higher MAR, and when hyper-tuned could become also the clear leader considering the other utilized metrics.

### 3. Not clear motivation for choosing the Acrobot system as the studied benchmark
As for the motivation of choosing the Acrobot system as the benchmark, it is not clear at all. Page 8 starts with the sentence _“Acrobot is a small but relatively difficult and fascinating system, so it is an ideal benchmark for
continuous-reward engineering systems”_. But I am not convinced, there is only a single other reference given providing the previous benchmark result K´egl et al. (2021). I would like to see more support for the claim that the Acrobot is the ideal system to study in the studied setting with adequate references.  A relation to classical optimal control theory would be a convincing example, and it could also serve as a baseline.

Nonetheless, even if the Acrobot system is an ideal benchmark it seems to be simplistic when compared to the usual continuous robotics RL benchmark environments studied in literature like Hopper or Half-Cheetah. I see significant differences between the Acrobot and the usual RL benchmark environments - the Acrobot does not require to deal with nonlinear forces generated by friction and contacts with the scene & other objects like in the other continuous RL benchmark envs.

### Other remarks:

* What is the ‘heating explorer’ used in the abstract,
* Please clarify the terminology used for the studied setting, I found in the paper and related literature several names, I assume, for the same problem:  iterated batch RL/ growing batch RL/ micro-data RL / micro-data model-based RL. I know that the field is new, but this naming discrepancy can become confusing in the long run.
* Please clarify what does precisely NaN reported in Table 2 is supposed to mean ?
* p.7, fig.3, l.5: I do not see how to interpret the notation with brackets T^{(i^*)}[1,2]?


**Summary Of The Paper:**

The authors study model-based reinforcement learning applied in the scenario of iterated batch reinforcement learning. There exist other namings of the studied in the paper scenario of iterated batch reinforcement learning, I found out that there exist several other synonymous names of such setting: growing batch RL/ micro-data RL / micro-data model-based RL. The authors motivate the problem informally using several bullet points listed in the introduction.

The main finding claimed by the authors is that the Dyna-style MBRL approach is an appropriate choice for the studied setting by enhancing it with decision-time planning based on a model-free policy and dynamically tuning exploration.

The algorithms are tested in the problem of resource-limited Acrobot; a detailed ablation study is provided for algorithm features (called here interchangeable algorithmic bricks for iterative batch RL. The performed ablation study helped to find the combination of a neural model, a bootstrapping DQN guide, and a heating explorer, which led to a 10% jump in state-of-the-art on the resource-limited Acrobot system.



**Summary Of The Review:**

I am on the fence regarding the significance of the paper results. On one hand, it adds new ideas and establishes an interesting line of research. Eventually has a significant potential of being useful in the research on MBRL.

However, the presentation of the paper is not mature enough. I think the paper has a few significant drawbacks including: lacks a proper mathematical formalization of the studied problem, and validation within a varied set of benchmarks and different scenarios. I explained my concerns in three paragraphs above. Therefore, I am leaning towards its rejection. An improved version of the paper may have a major impact.

More adequate score is 4.5

---

> ### Author Response · Authors · 2021-11-22
> **Response to reviewer jVBs**
>
> We first would like to thank the reviewer for the constructive feedbacks and comments.
>
> **After reading the studied problem statement in the introduction, it is still unclear for me what precisely can be characterized using the term ‘iterated batch RL’ coined by the authors. In particular, why it is necessary to define and name an MBRL study subgenre, rather than use the existing framework of online/offline MBRL.**.
>
> "Online RL" assumes that interacting with the real system is possible at any-time and that the policy can be updated after each single-step interaction. In the "offline RL" setting (also known as "batch RL") the agent is not allowed to interact with the system and only a static dataset can be used to learn a policy. Our setting is neither offline RL, as we assume that we can interact with the system during short period of time, nor online RL, as the policy cannot be updated during these interaction periods. The "iterated batch RL" term represents well the fact that the batch RL setting is repeated, each interaction period resulting in a larger and more diverse dataset to update your system model and policy. One of the main differences with batch RL is that iterated batch RL can benefit from exploration on the real system. Iterated batch is often called growing batch but we don't like the terms since the training data doesn't simply grow uncontrolably, but generated by the very iterative process of collection/training.
>
> We clarified this in the introduction.
>
> We cannot use the term offline MBRL as this is commonly used to denote a setting with no future system interaction to improve the policy (see e.g. Argenson  and Dulac-Arnold (2020)). We agree with Levine et al. (2020) when they say that the common use of the term "offline/batch RL" to describe a setting where no future interaction with the system is allowed can lead to confusion. This is why we refer to it as _pure_ offline/batch.
>
> **However, the 10% improvement may be the result of a simple hyperparameter optimization.  Therefore, it is not unexpected that an exhaustive hyper-parameter search could significantly boost the MAR/MRCP scores of the other evaluated algorithms and change the final leaderboard. Table 3 given in the appendix indicates that the authors did not, in fact, perform an exhaustive hyperparameter search.**
>
> Could the reviewer explain why Table 3 would indicate that we did not perform an exhaustive hyperparameter search?
> Hyperparameter spaces of RL algorithms are very large. This is why we optimized the agent independently of the models. For the DQN policy we tried a combination of different parameters for the epsilon greedy strategy, the discount factor gamma, the batch size, the length of the replay buffer, the number of steps before resetting to a previous real state and the NN architecture for the Q values etc... We used and reported the one leading to the best performance. It is possible that another set of hyperparameters would give a better result for DQN alone but the gap between DQN alone and DQN with planning remains large.
>
> **Another concern is related to the fact that the DynaZero benchmark algorithm achieves higher MAR, and when hyper-tuned could become also the clear leader considering the other utilized metrics.**
>
> We also carefully tuned the DynaZero agent. We actually had a hard a time tuning it so that it achieves a performance similar to the other well-performing agents.
>
> **Not clear motivation for choosing the Acrobot system as the studied benchmark**
>
> Acrobot is a small yet not so simple system. It allows to run extensive comparisons at an acceptable cost. Designing affordable extensive experimental studies is important to make the RL research more inclusive (see e.g. "Revisiting Rainbow: Promoting more insightful and inclusive deep reinforcement learning research" by Obando-Ceron and Castro (2021)) and ease reproducibility.
>
> **What is the ‘heating explorer’ used in the abstract?**
> The heating explorer is detailled in Figure 4 "HeatingExplore": it heats the guide action distribution to n different temperatures, one per action sequence.
>
>
> **I do not see how to interpret the notation with brackets $T^{(i^\*)}[1,2]$**
>
> $T^{(i^*)}[1,2]: T^{(i^*)}$ is the best trace. Each trace is a 2D list of (state, action) pairs, so $T[1, 2]$ is the action of the first element of the trace. So $T^{(i^*)}[1,2]$ is the "first action of the best trace", as the comment  says in the pseudocode. It's a computer-sciency notation, but it's completely formal. We added a sentence in the caption to make this clear.

---

> > ### Comment · Reviewer_jVBs · 2021-11-29
> > **thanks for the reply**
> >
> > Thanks for your reply and taking time to revise the paper with additional clarifications. At this point I will keep my score, as the paper needs major revision, especially taking into account suggestions that were common for the other reviews.  I mean in particular
> > 1. The crucial statements, like the iterated batch RL vs model-based RL require formal mathematical definitions to be clear about why the studied setting is different, other examples of terms requiring necessary definitions were given by the other reviewers.
> > 2. The algorithm needs to be tested in wider set of scenarios and more complicated environments , especially those including other external forces coming from friction and ground contact, the same remark was raised by the other reviewers.
> > 3. By an exhaustive hyper-parameter search I mean checking more values in the studied parameter intervals preferably using techniques like random or Bayesian optimization.
> > 4. If you tuned also the hyper-parameters of DynaZero please also include a table with the tested hyperparameters.

---

### Official Review · Reviewer_H38y · 2021-11-05

**Correctness:** 3
**Technical Novelty And Significance:** 1
**Empirical Novelty And Significance:** 2
**Recommendation:** 3
**Confidence:** 2

**Main Review:**

Strengths
1. The iterated batch RL motivation is interesting
2. The scope of ablations is above average

Weaknesses
1. While I find the first paragraph or two of the introduction to be enjoyable, the informal, chatty approach to writing the rest of the paper detracts from the content. As an example: "Our main contribution is a Dyna-style GUIDE&EXPLORE strategy (Fig 2). The gist is to learn a guide policy ξ using a model-free RL technique on the model p and on the traces collected on the real system T". This is the main contribution! What is a guide, what is Dyna-style, why are we talking about a "gist"? Ordinarily I would avoid commenting on writing style in a review, but it must be addressed.

2. Further, if we are to understand the above comment as the main contribution, the issue is that "the guide" is a concept, not a new concept at that (just a renaming), and this paper doesn't clearly state a hypothesis for any particular implementation.  The paper spends a section addressing terminology in the description of the guide we are told that sometimes the guide is trained on data from the model of the system (mostly in fact), but also sometimes ("when possible") trained with off policy data.

3. Again in the "terminology section" the following appears: "In that sense, through the data from the **real system**, planning is part of training the guide". It appears that here the paper refers to the OpenAI Gym as the **real system**, the paper should clarify that a better experimental setup would include a real, real system, a simulator of the real system, and a learned model of the real system. Others have done physical acrobot experiments (c.f. https://arxiv.org/pdf/1210.0888.pdf, http://youtu.be/FeCwtvrD76I)

4. Why do NaNs appear in some of the tables?

5. What is the contribution? I see the list at the end of the introduction, but none are testable/falsifiable statements.

**Summary Of The Paper:**

The paper describes meta-method and metrics for solving iterated batch reinforcement learning problems. The paper concludes with experiments on an "acrobot" simulation.

**Summary Of The Review:**

This paper explored a variety of approaches for iterated batch reinforcement learning. There is no obvious contribution other than the demonstration that experimentally both planning and exploration are important to improving the performance of Acrobot policies.

---

> ### Author Response · Authors · 2021-11-22
> **Response to reviewer H38y**
>
> We would greatly appreciate if the reviewer could elaborate on what we could improve. For instance when he writes "What is a guide, what is Dyna-style..." Does the reviewer suggest that all this would need to be better defined by the time we state our contributions in the introduction? Or that is is unclear after reading the paper?

---

> > ### Comment · Reviewer_H38y · 2021-11-29
> > **Refining my question**
> >
> > Both:
> >
> > 1. The "guide" and "dyna-style" concepts should be defined without jargon. The reason is that it is hard to tell if they are new ideas. The paper states that "guide" is a renaming of an existing concept. "Dyna-style" is also an existing concept. Restating the contribution without jargon would clarify how this work is different. So yes, before the claim of a contribution is made it should be clear what the claim is.
> >
> > 2. Secondly, much of the hypothesizing in this paper doesn't occur until "after results are known". "Dyna-style" and "guide" terms are used as placeholders for design choices which are not disambiguated until after the experiments are performed. If the contribution is a framework (I think this is the intent) the positioning/advice/takeaway given the structure of the paper lacks clarity and supporting evidence. As far as I can tell the paper doesn't demonstrate that dyna-style guide and explore strategies dominate all other methods. Rather, that some specific versions of the approach on a not completely standard instance of Acrobot do better when analyzed with, admittedly, possibly better, but still non-standard metrics.
> >
> > There does not seem to have been significant edits to the draft addressing these comments or similar suggestions by other reviewers (e.g. jVBs, which were not considered too ambiguous to respond to directly). Thus, I am leaving my score unchanged at a low confidence to allow other reviewers to influence the decision if they feel strongly.

---

### Author Response · Authors · 2021-11-22
**General response**

We thank the reviewers for their feedbacks. We address common issues in this general response.

**Lack of formality:** We are somewhat surprised by the reviewers' complaint on lack of formality since we are actually _proud_ of the concise and mathematically rigorous description of all what we do. It would help a lot in improving the paper if you could specify what algorithmic detail is unclear. We would be immensely grateful.

**NaNs in the tables:**
The NaN for the MAR standard deviation of DynaZero in Table 1 is a typo and has been fixed.
The NaNs for the MRCP results of DQN and PPO (Table 2 and Table 3) mean that these agents do not reach the reward threshold (1.8) used to compute the MRCP.
The NaN for the MRCP standard deviation of DynaZero in Table 2 is because we only ran 1 seed for this agent as it is very time consuming.

---

> ### Comment · Reviewer_EmNA · 2021-11-23
> **lack of formality**
>
> Thank you for your responses and clarifications.  I can't speak for all of the reviewers but from my perspective and reading the other reviews I see value in he experiment conducted here but we need to see more domains / settings.  On the formalism question, I think there is concern about (1) the informal examples in the introduction, which don't have grounding in the paper, (2) terms being used without definitions as "H38y" gave concrete examples for and i mentioned in the exploration section, and (3) there was no formal hypothesis stated at the beginning and no theoretical derivations.  I think adding a more rigorous problem definition section and expanding the experiments to other domains would make this a much stronger paper.

---

### Decision · Program_Chairs · 2022-01-20

**Decision:**

Reject

**Comment:**

The paper studies dyna-style MBRL in a resource-limited setting. It is evaluated on an acrobat task where it shows very promising results.

The reviewers appreciated the extensive replies, but they did not fundamentally change their opinion. In particular:
- Lack of formal problem statement and definitions
- The experiment on a single task (and that being a non-standard version) isn't sufficient to demonstrate the general merits of the method

While the ideas are very promising, the paper cannot be published in its current form. We'd hence like to highly encourage the authors to revise the paper and to re-submit at a different venue.